# A Pediatric Case of *COLQ*-Related Congenital Myasthenic Syndrome with Marked Fatigue

**DOI:** 10.3390/children10050769

**Published:** 2023-04-24

**Authors:** Takuya Horibe, Hideki Shimomura, Sachi Tokunaga, Naoko Taniguchi, Tomoko Lee, Shigemi Kimura, Yasuhiro Takeshima

**Affiliations:** 1Department of Pediatrics, Hyogo Medical University School of Medicine, Nishinomiya 663-8501, Japan; horibe.hyo@gmail.com (T.H.); sa-tokunaga@hyo-med.ac.jp (S.T.); nao-taniguchi@hyo-med.ac.jp (N.T.); leeleetomo@me.com (T.L.); ytake@hyo-med.ac.jp (Y.T.); 2Children’s Rehabilitation, Sleep and Development Medical Center, Hyogo Prefectural Rehabilitation Central Hospital, Kobe 651-2134, Japan; 3658kimura1@gmail.com

**Keywords:** congenital myasthenic syndrome, collagen-like tail subunit of asymmetric acetylcholinesterase, fatigue

## Abstract

Congenital myasthenic syndrome (CMS) is a clinically and genetically heterogeneous inherited disorder that is treatable. Although the disease usually develops at birth or during infancy, some patients develop the disease in the second to third decades of life. Collagen-like tail subunit of asymmetric acetylcholinesterase (*COLQ*)-related CMS is CMS with mutations in the *COLQ*, which results in end-plate acetylcholinesterase deficiency. Diagnostic delay is common in patients with later-onset CMS due to slow progression and fluctuating symptoms. Understanding CMS with atypical and unusual presentations is important to treat this condition effectively. Here, we report a case of *COLQ*-related CMS. A 10-year-old girl presented with only marked fatigue, which was provoked by exercise but improved after 30–60 min of rest. While motor nerve conduction velocity was normal, a compound muscle action potential (CMAP) with four peaks was recorded. Repetitive stimulation of the accessory nerve exhibited a decrease in CMAP amplitude. Genetic tests revealed compound heterozygous mutations in *COLQ* (c.1196-1_1197delinsTG and c.1354C>T). Treatment with salbutamol improved fatigue but not the electrophysiological markers. Thus, significant fatigue is a hallmark of *COLQ*-related CMS; early diagnosis is essential for ensuring appropriate treatment.

## 1. Introduction

Congenital myasthenic syndromes (CMS) are rare genetic disorders comprising a subset of neuromuscular disorders in which genetic defects result in the dysfunction of proteins comprising the neuromuscular junction (NMJ) [1]. Collectively, these proteins participate in the structure, function, and repair of the NMJ [1]. As a clinically and genetically heterogeneous set of diseases [1], the clinical spectrum of CMS is highly variable, ranging from minor symptoms to progressive disabling weakness [2]. In some subtypes of CMS, acute worsening of symptoms is triggered by infection, fever, or exercise [3].

Some patients with CMS develop non-specific clinical manifestations, such as bulbar and respiratory difficulties, arthrogryposis, delayed motor development, and progressive respiratory or bulbar weakness [1]. Therefore, diagnosing CMS is often challenging, particularly during childhood [4]. Consequently, children are sometimes misdiagnosed with other neuromuscular disorders, which leads to treatment delays and unnecessary examinations [5]. Previous studies have shown that approximately 80% of pediatric patients are misdiagnosed [6]. The reason for this is assumed to not only be that CMS presents non-specific symptoms, but also that symptoms differ according to age. Patients who develop in early childhood show motor delay without other symptoms; however, along with age, the manifestations characteristic of CMS, which are muscle weakness with diurnal fluctuations and fatigue, become apparent [7].

Collagen-like tail subunit of asymmetric acetylcholinesterase (*COLQ*)-related CMS is a type of CMS with mutations in the *COLQ*, resulting in end-plate acetylcholinesterase (AChE) deficiency. Most patients with this subtype are severely disabled from an early age with respiratory difficulties and progressive involvement of the axial muscles, leading to severe scoliosis and restrictive ventilatory deficits [7]. Although CMS develops via a complicated mechanism, drug treatment with β-adrenergic agonists is known to be remarkably effective in *COLQ*-related CMS [1]. Here, we report a pediatric case of *COLQ*-related CMS manifesting as marked fatigue that was successfully treated.

## 2. Case Presentation

A 10-year-old girl with difficulty walking was referred to our hospital. She had nonconsanguineous parentage and normal developmental milestones. At the age of 9 years, she began to have difficulty running quickly and complained of fatiguability; these complaints had progressively worsened in the year leading up to her visit to our hospital. The symptoms were provoked by exercise, though fatigue improved after 30–60 min of rest. On physical examination at rest, manual muscle testing and deep tendon reflexes were normal. No ptosis, ophthalmoplegia, facial weakness, dysphagia, or skeletal deformities were noted. However, when both upper limbs were raised to shoulder height and held in place, her upper limbs dropped down to her side within 5 s. In another exercise involving stair-climbing, a waddling gait appeared after approximately four steps. However, after a short rest, she was able to resume her usual activities.

Serum levels of creatine kinase, aspartate transaminase, and lactate dehydrogenase were normal. Tests for serum antibodies to the acetylcholine receptor and muscle-specific tyrosine kinase were negative. Brain magnetic resonance imaging was normal. Motor nerve conduction velocity with a single stimulation in the median nerve was 50.6 m/s, and a compound muscle action potential (CMAP) with four peaks, consistent with a repetitive CMAP (R-CMAP), was recorded (Figure 1). Repetitive nerve stimulation (RNS) in the accessory nerve with 5 Hz stimulation exhibited a 28.8% decrease in CMAP amplitude at the fourth stimulus (Figure 2), a pattern that was not observed in the median nerve.

Next-generation sequencing of the patient’s genomic DNA revealed two heterozygous mutations in the *COLQ* gene: a heterozygous mutation at the splice site c.1196-1_1197delinsTG and a heterozygous missense mutation c.1354C>T. Genetic testing of her parents showed that the former mutation was present in the father, and the latter mutation was present in the mother. Thus, based on these pathogenic mutations, it was considered a compound heterozygous mutation. The c.1354C>T mutation has been previously reported in affected patients [8]. To understand the clinical significance of the splice site mutation, RNA was isolated from peripheral blood leukocytes, and mRNA analysis was performed. Analysis revealed a product with exon 16 skipped, and the mutation was considered to be pathogenic (Figure 3). Depending on the guideline for sequence variants in the American College of Medical Genetics, the variant c.1196-1_1197delinsTG (p.Arg399delins = fs * 17) was satisfied with the categories PVS1 and PM2, which are categorized as class 4 (likely pathogenic). The variant c.1354C>T (p.Arg452Cys) was satisfied with PS1, PM2, PP3, and PP4, which were also categorized as class 4 (likely pathogenic).

The patient was diagnosed with *COLQ*-related CMS and administered salbutamol 8 mg/day, which improved her fatigue markedly, such that she could raise her upper limbs for 27 s and climb four flights of stairs without waddling gait or fatigue. Repeat electrophysiological examination after 6 months of daily salbutamol treatment still showed R-CMAP and CMAP amplitude decreases with stimulation of the accessory nerve. Examination of the median nerve revealed a CMAP decrease of 14.7% in amplitude.

## 3. Discussion

Although most CMS cases can be treated with drugs, unfortunately a delayed diagnosis can affect prognosis [9]. The rate of misdiagnosis in CMS is high due to non-specific and fluctuating presenting symptoms [6,10,11]. The initial manifestation in this patient was only marked fatigue. Since CMS is a rare disease, it was difficult to diagnose; however, electrophysiological examination was helpful in establishing the diagnosis. Fortunately, the administration of a β-adrenergic agonist showed a beneficial effect, and the patient’s activity in daily life improved dramatically with treatment.

Patients with CMS typically present with a history of fatigable weakness involving the ocular, bulbar, and limb muscles, with onset shortly after birth or in early childhood, usually within the first two years [3]. Delayed motor developmental milestones are often observed in patients with early onset [7]. However, some patients present with other manifestations in the second to third decades of life [12]. Individuals with onset in later childhood exhibit abnormal muscle fatigue, with fluctuation in symptoms over various periods [7,10]. The patient in this report presented with marked fatigue, with symptoms provoked by exercise lasting for 30–60 min. A previous study reported that all patients with *COLQ*-related CMS experienced daytime fluctuations of symptoms that were highly dependent on exercise [13]. For instance, there have been two episodes of tetraparesis in a 17-year-old patient 1 h after rambling [13].

Fatigue is a common manifestation of various neuromuscular diseases, including multiple sclerosis, Parkinson’s disease, myasthenia gravis, CMS, and dermatomyositis [6]. Although the term “fatigue” has been used to refer to a symptom of neurological disease, there is no standard definition [6]. However, exercise intolerance, which is used as a term with a similar meaning to “fatigue,” has been known to be a major symptom of metabolic disorders [14]. Exercise intolerance induced by short bursts of high-intensity exertion, usually within approximately 5 min, should lead clinicians to consider glycogen storage disorders [14]. Precipitation by prolonged periods of low-intensity (endurance) exercise is more typical of fatty acid oxidation and mitochondrial disorders [14]. Since the symptoms in this case were induced by short bursts of high-intensity exertion, glycogen storage disorder type 5, McArdle disease, was initially suspected. However, in this patient, there were no major features associated with glycogen storage disorder, such as post-exertional pigmenturia, rhabdomyolysis, or hyper creatine-kinase-emia.

With respect to electrophysiological examinations, the diagnosis of CMS due to R-CMAP was considered in this case. R-CMAP is a CMAP with polyphasic discharge, which is attributed to repeated muscle fiber discharge after a single electric stimulation [15]. A previous study reported that the special electrophysiological manifestations of *COLQ*-related CMS are under-recognized [16]. R-CMAP has been observed in two types of CMS: end-plate AChE deficiency and slow-channel syndrome [11]. Another electrophysiological finding in CMS is a decrease in CMAP during RNS [17,18]. However, this is not a CMS-specific finding and is common in myasthenia gravis [17]. Despite improvement in fatigue in this case, neither electrophysiological finding improved after drug treatment with a β-adrenergic agonist. Thus, it is suggested that this drug did not improve the underlying function.

To date, many genetic mutations associated with CMS have been identified, but the phenotype–genotype correlation is not well known [7,19]. Approximately half of CMS cases are caused by genetic defects in the *CHRNE* gene, which encodes the acetylcholine receptor epsilon (ε) subunit. The second most common mutation is in the *RAPSN* gene, which encodes a receptor-associated protein of the synapse, contributing to 15–20% of CMS cases. Other frequently mutated genes include *DOK7* and *COLQ*, both accounting for 10–15% of CMS cases [19]. The *COLQ* gene, encoding the collagen-like tail subunit of asymmetric AChE, is responsible for AChE deficiency. This is caused by biallelic loss-of-function mutations in the *COLQ* gene [10]. COLQ plays an important structural role in the NMJ by anchoring AChE to the basal lamina and accumulating AChE in NMJ [12]. Therefore, *COLQ* mutations result in AChE deficiency, extending the lifetime of acetylcholine in the synaptic space [7]. This results in prolonged exposure of cholinergic receptors to acetylcholine, causing receptor desensitization and end-plate myopathy due to cationic overloading of the postsynaptic region [20]. Genetic testing in this patient revealed compound heterozygous mutations in the splice site c.1196-1_1197delinsTG and missense mutation c.1354C>T in the *COLQ* gene, the latter of which has been reported previously [8].

Although the underlying mechanism of *COLQ*-related CMS has been elucidated, a specific therapeutic drug has yet to be developed. However, the β-adrenergic agonists salbutamol and ephedrine are effective drugs for *COLQ*-related CMS [1]. The mechanism of action of these drugs includes various effects of regulating skeletal muscle structure and function, which promotes an anabolic effect in the metabolism of skeletal muscle [21]. In this report, improvements in several postsynaptic morphological defects such as increased synaptic area, acetylcholine receptor area and density, and extent of postjunctional folds are considered as the assumed mechanism [21]. An experiment with rats reported that these actions are predominantly mediated through β2 receptors and involve cAMP signaling [22]. An AChE inhibitor, usually used in patients with myasthenia gravis, should not be used for the diagnosis and drug treatment in *COLQ*-related CMS. This is because AChE inhibitors increase the accumulation of AChE in the NMJ, resulting in worsened muscle weakness.

Diagnosing *COLQ*-related CMS can often be difficult, or even missed, owing to non-specific symptoms. Episodes of marked fatigue and R-CMAPs can be diagnostic clues to *COLQ*-related CMS. The efficacy of drug therapy should be assessed based on clinical symptoms, and the assessment of prognosis in electrophysiological examinations is needed for future studies.

## Figures and Tables

**Figure 1 children-10-00769-f001:**
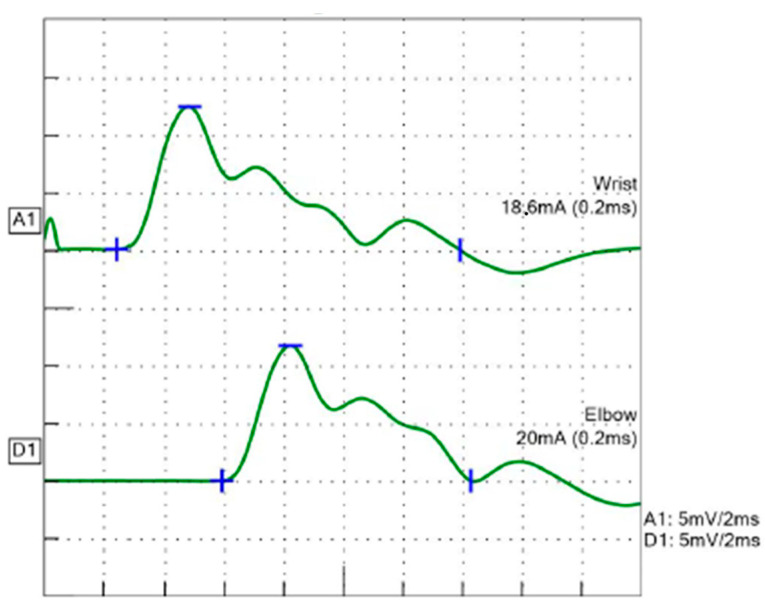
Motor nerve conduction velocity in the median nerve. Four CMAPs were recorded. Blue lines indicated the start point (+), maximum value (−), and end point (|) of the waveform, respectively.

**Figure 2 children-10-00769-f002:**
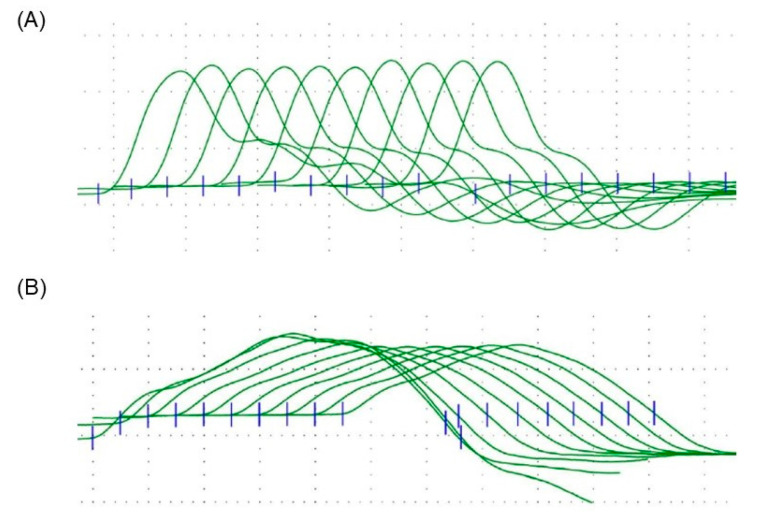
Repetitive nerve stimulation studies with 5 Hz frequencies in the accessory and median nerves. Blue lines indicated the start point of each wave. (**A**) Median nerve study showed no decremental response. (**B**) Accessory nerve showed a decremental response of 28.8%.

**Figure 3 children-10-00769-f003:**
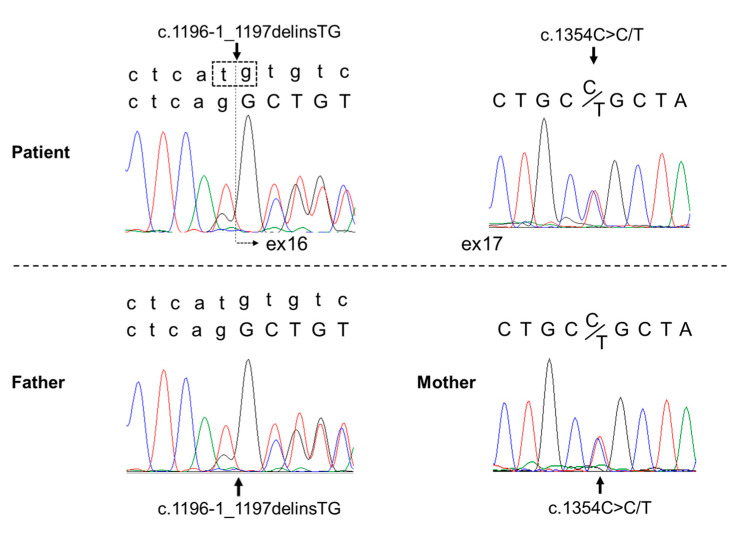
Genetic examination in the patient and her biological parents. In the two-row array shown in the patient’s left schema and the father’s schema, the upper line is the pathogenic sequence, and the bottom line is the normal sequence. Compound heterozygous mutations were found in the patient’s *COLQ* gene: the splicing mutation c.1196-1_1197delinsTG and the missense mutation c.1354C>T. Colored curves indicate each base detected by the Sanger sequencing.

## Data Availability

The data that support the findings of this study are available from the corresponding author upon reasonable request.

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
