# Peer review of "A Pediatric Case of COLQ-Related Congenital Myasthenic Syndrome with Marked Fatigue"

_children, 2023, doi:10.3390/children10050769_

Round 1

Reviewer 1 Report

1. I suggest authors to include a brief explanation of why do salbutamol, albuterol or ephedrine promotes marked improvement of symptoms. 

2. Were sleep studies performed during diagnostic work-up (i.e., polysomnography)? 

3. Authors should review the manuscript to provide all mention to COLQ gene in italics, including the "COLQ-related CMS".

4. A minor suggestion for authors is to present the current classification of pathogenicity of the observed variants, according to the ACMG 2015 criteria: 

- c.1196-1_1197delinsTG (p.Arg399delins???): class 4 - probably pathogenic; PVS1 and PM2 criteria. 

- c.1354C>T (p.Arg452Cys): class 3 - variant of unknown significance (VUS); PM2 and PP3 criteria. 

Author Response

We appreciate the comments and suggestions from the expert Reviewers, which helped us enhance the quality of our manuscript. We have addressed your comments with point-by-point responses listed below.

1. I suggest authors to include a brief explanation of why do salbutamol, albuterol or ephedrine promotes marked improvement of symptoms. 

Answer: We appreciate the reviewer's comment regarding additional treatment information, and have added following description to the Discussion section.

Line 169

In this report, improvements in several postsynaptic morphological defects such as in-creased synaptic area, acetylcholine receptor area and density, and extent of postjunctional folds are considered as the assumed mechanism [21].

2. Were sleep studies performed during diagnostic work-up (i.e., polysomnography)? 

Answer: Thank you for your comment. However, polysomnography was not conducted in this study.

3. Authors should review the manuscript to provide all mention to COLQgene in italics, including the "COLQ-related CMS".

Answer: Thank you for your suggestion. We altered these descriptions.

4. A minor suggestion for authors is to present the current classification of pathogenicity of the observed variants, according to the ACMG 2015 criteria: 

- c.1196-1_1197delinsTG (p.Arg399delins???): class 4 - probably pathogenic; PVS1 and PM2 criteria. 

- c.1354C>T (p.Arg452Cys): class 3 - variant of unknown significance (VUS); PM2 and PP3 criteria. 

Answer: Thank you for your important suggestion. Consistent with your comment, we considered c.1196-1_1197delinsTG (p.Arg399delins???) was likely pathogenic; PVS1 and PM2 criteria. However, c.1354C>T (p.Arg452Cys) which satisfied PS1, PM2, PP3, and PP4 was considered to be likely pathogenic. To address it, we have altered these descriptions as follows.

Line 86

Depend on the guideline for sequence variants in American College of Medical Genetics, the variants, c.1196-1_1197delinsTG (p.Arg399delins=fs*17) were satisfied with the categories, PVS1 and PM2, which categorized as class 4 (likely pathogenic). The variant c.1354C>T (p.Arg452Cys) was satisfied with PS1, PM2, PP3, and PP4, which were also categorized as class 4 (likely pathogenic).

Additional correction.

We found an error in approval number of Institutional Review Board Statement. Thus, we have changed these descriptions as follows.

This study was approved by the Review Board of Hyogo Medical University (approval no. rinhi-63).

Reviewer 2 Report

This paper is well-written, with clear presentation of CMS. I would like to recommed this case report to be accepted after some revision.

1.   Fatigue and  fatigability. Although you have explained the definition of  fatigability in the discusion part, however, (https://journals.lww.com/acsm-essr/fulltext/2016/10000/fatigue_and_fatigability_in_persons_with_multiple.1.aspx) : fatigue is characterized as a symptom and should be distinguished from fatigability, which indicates how quickly a specific level of fatigue is achieved. In addition, (https://pubmed.ncbi.nlm.nih.gov/34583577/) also clearly define fatigability as estimated (perceived fatigability) or measured (objective fatigability);  Collins dictionary defines fatigability as the quality of being susceptible to fatigue. In sum, all those mentioned above suggest that fatigability is something needed to be “measured”. Therefore, I feel that this term should not be used as a complaint.

2.       in your discussion, line 150 “This is caused by biallelic loss-of- function mutations in the COLQ gene”. What does this mean?

3.      In your figure 3 patient’s sequences, c.1196-1_1197delinsTG  this one is only “one allele”. How do you explain? I may contribute your case to an assumption that one allele mutation (heterozygosity) may not present any symptoms (This may explain the patient’s parents have no symptoms). However, double mutations may cause clinical presentations.

4.      Line 118: For instance, there have been two episodes of tetraparesis in an 18-year-old patient 1 h after rambling. The paper shows 17-year-old.

5.      Line 128-130 glycogen storage disorder was initially suspected. However, in this patient, there were no major features associated with glycogen storage disorder, such as post-exertional pigmenturia, rhabdomyolysis, or hyper creatine-kinase-emia.

Normally, most clinicians may consider the diagnosis of GSD if any patients present doll-like faces with fat cheeks, relatively thin extremities, short stature, and protuberant abdomen, and xanthomas. I suggest you to add these phenotype presentations of GSD in your discussion rather than post-exertional pigmenturia,  rhabdomyolysis, or hyper creatine-kinase-emia

6.      Line 137-138 Another electrophysiological finding in CMS is a decrease in CMAP during RNS [1].

I went through the reference (1), found that the original references for this finding is

(a.)  Janas JS, Barohn RJ. A clinical approach to the congenital myasthenic syndromes. J Child Neurol. 1995;10(2):168–9.

(b.)Abicht A, Müller JS, Lochmüller H. GeneReviews®: Congenital myasthenic syndromes. Seattle (WA): University of Washington, Seattle; 2003 May 9

     (c.) Harper CM, Engel AG. Quinidine sulfate therapy for the slow-channel congenital myasthenic syndrome. Ann Neurol. 1998;43(4):480–4

Author Response

We appreciate the comments and suggestions from the expert Reviewers, which helped us enhance the quality of our manuscript. We have addressed your comments with point-by-point responses listed below.

1. Fatigue and  fatigability. Although you have explained the definition of  fatigability in the discusion part, however, (https://journals.lww.com/acsm-essr/fulltext/2016/10000/fatigue_and_fatigability_in_persons_with_multiple.1.aspx) : fatigue is characterized as a symptom and should be distinguished from fatigability, which indicates how quickly a specific level of fatigue is achieved. In addition, (https://pubmed.ncbi.nlm.nih.gov/34583577/) also clearly define fatigability as estimated (perceived fatigability) or measured (objective fatigability);  Collins dictionary defines fatigability as the quality of being susceptible to fatigue. In sum, all those mentioned above suggest that fatigability is something needed to be “measured”. Therefore, I feel that this term should not be used as a complaint.

Answer: Thank you for your important comment. To address your comment, we have altered the term “fatigability” to “fatigue”. The term we used to describe "fatigability" in the discussion section was meant to be "fatigue".

2. in your discussion, line 150 “This is caused by biallelic loss-of- function mutations in the COLQ gene”. What does this mean?

Answer: Thank you for your comment. I have added a reference number 10 to this article as it was missing.

This is caused by biallelic loss-of-function mutations in the COLQ gene [10].

3. In your figure 3 patient’s sequences, c.1196-1_1197delinsTG  this one is only “one allele”. How do you explain? I may contribute your case to an assumption that one allele mutation (heterozygosity) may not present any symptoms (This may explain the patient’s parents have no symptoms). However, double mutations may cause clinical presentations.

Answer: Thank you for your comment. In this case, we thought that compound heterozygous mutations occurred due to the presence of pathogenic mutations in both her father and mother. To address your concerns, we added the description as follows.

Line 81

Thus, based on these pathogenic mutations it was considered a compound heterozygous mutation.

Line86

Depend on the guideline for sequence variants in American College of Medical Genetics, the variants, c.1196-1_1197delinsTG (p.Arg399delins=fs*17) were satisfied with the categories, PVS1 and PM2, which categorized as class 4 (likely pathogenic). The variant c.1354C>T (p.Arg452Cys) was satisfied with PS1, PM2, PP3, and PP4, which were also categorized as class 4 (likely pathogenic).

4. Line 118:For instance, there have been two episodes of tetraparesis in an 18-year-old patient 1 h after rambling. The paper shows 17-year-old.

Answer: Thank you for pointing out this error which has been corrected in the revised manuscript into 17-year-old.

5. Line 128-130 glycogen storage disorder was initially suspected. However, in this patient, there were no major features associated with glycogen storage disorder, such as post-exertional pigmenturia, rhabdomyolysis, or hyper creatine-kinase-emia.

Normally, most clinicians may consider the diagnosis of GSD if any patients present doll-like faces with fat cheeks, relatively thin extremities, short stature, and protuberant abdomen, and xanthomas. I suggest you to add these phenotype presentations of GSD in your discussion rather than post-exertional pigmenturia, rhabdomyolysis, or hyper creatine-kinase-emia

Answer: Thank you for your important comment. The vague description caused some confusion. We initially suspected glycogen storage disease type 5, McArdle disease, which manifests post-exertional pigmenturia, rhabdomyolysis, or hyper creatine-kinase-emia. To state that clearly, we added description as follows.

Line 134

Since the symptoms in this case were induced by short bursts of high-intensity exertion, glycogen storage disorder type 5, McArdle disease, was initially suspected.

6. Line 137-138 Another electrophysiological finding in CMS is a decrease in CMAP during RNS [1].

I went through the reference (1), found that the original references for this finding is

(a.)  Janas JS, Barohn RJ. A clinical approach to the congenital myasthenic syndromes. J Child Neurol. 1995;10(2):168–9.

(b.)Abicht A, Müller JS, Lochmüller H. GeneReviews®: Congenital myasthenic syndromes. Seattle (WA): University of Washington, Seattle; 2003 May 9

 (c.) Harper CM, Engel AG. Quinidine sulfate therapy for the slow-channel congenital myasthenic syndrome. Ann Neurol. 1998;43(4):480–4

Answer: Thank you for your suggestion. As you suggested, I changed them to the appropriate citations.

Additional correction.

We found an error in approval number of Institutional Review Board Statement. Thus, we have changed these descriptions as follows.

This study was approved by the Review Board of Hyogo Medical University (approval no. rinhi-63).
